# Inhibition of Rho-Associated Kinase Suppresses Medulloblastoma Growth

**DOI:** 10.3390/cancers12010073

**Published:** 2019-12-26

**Authors:** Cecilia Dyberg, Teodora Andonova, Thale Kristin Olsen, Bertha Brodin, Marcel Kool, Per Kogner, John Inge Johnsen, Malin Wickström

**Affiliations:** 1Childhood Cancer Research Unit, Department of Women’s and Children’s Health, Karolinska Institutet, 171 77 Stockholm, Sweden; cecilia.dyberg@ki.se (C.D.); teodora.andonova@ki.se (T.A.); thale.kristin.olsen@ki.se (T.K.O.); per.kogner@ki.se (P.K.); john.inge.johnsen@ki.se (J.I.J.); 2Department of Microbiology Tumor and Cell Biology (MTC), Karolinska Institutet, 171 77 Stockholm, Sweden; bertha.brodin@ki.se; 3Hopp Children’s Cancer Center (KiTZ), 61 120 Heidelberg, Germany; m.kool@kitz-heidelberg.de; 4Division of Pediatric Neuro-Oncology, German Cancer Research Center (DKFZ) and German Cancer Consortium (DKTK), 61 120 Heidelberg, Germany; 5Princess Máxima Center for Pediatric Oncology, 3584 CS Utrecht, The Netherlands

**Keywords:** medulloblastoma, metastasis, Rho, ROCK, Rho-associated kinase, ROCK inhibitor, epithelial-mesenchymal transition

## Abstract

Medulloblastoma is one of the most common malignant brain tumor types in children, with an overall survival of 70%. Mortality is associated with metastatic relapsed tumors. Rho-associated kinases (ROCKs), important for epithelial-mesenchymal transition (EMT) and proper nervous system development, have previously been identified as a promising drug target to inhibit cancer growth and metastatic spread. Here, we show that ROCKs are expressed in medulloblastoma, with higher ROCK2 mRNA expression in metastatic compared to non-metastatic tumors. By evaluating three ROCK inhibitors in a panel of medulloblastoma cell lines we demonstrated that medulloblastoma cells were sensitive for pharmacological ROCK inhibition. The specific ROCK inhibitor RKI-1447 inhibited the tumorigenicity in medulloblastoma cells as well as impeded cell migration and invasion. Differential gene expression analysis suggested that ROCK inhibition was associated with the downregulation of signaling pathways important in proliferation and metastasis e.g., TNFα via NFκβ, TGFβ, and EMT. Expression of key proteins in these pathways such as RHOA, RHOB, JUN, and vimentin was downregulated in ROCK inhibited cells. Finally, we showed that ROCK inhibition by RKI-1447 suppressed medulloblastoma growth and proliferation in vivo. Collectively, our results suggest that ROCK inhibition presents a potential new therapeutic option in medulloblastoma, especially for children with metastatic disease.

## 1. Introduction

Medulloblastoma is one of the most common malignant childhood brain tumors. Current therapies include surgery, radiotherapy, and chemotherapy. The overall 5-year survival rate is about 70%, however, survivors often suffer from permanent neurocognitive sequelae. Medulloblastoma arises in the cerebellum and is believed to originate from neuronal progenitor cell populations during early life. Genetic aberrations in important genes during early embryogenesis are believed to be initiating events. Gene expression profiling has divided medulloblastoma into at least four distinct molecular subgroups including Wingless (Wnt), Sonic Hedgehog (Shh), Group 3 and Group 4. These subgroups exhibit specific molecular drivers and are associated with different clinical features and outcomes [1,2]. Medulloblastoma patients seldom die due to the primary tumor or local recurrence, but rather because of metastatic spread at the time of relapse [3,4]. The HIT-SIOP-PNET4 study showed an overall survival of 6% 5 years after relapse [5]. Medulloblastoma metastases are found almost exclusively on the leptomeningeal surface of the brain and spinal cord, which has led to the assumption that medulloblastoma spreads through the cerebrospinal space. Recent evidence suggests that it also can spread through circulating tumor cells in the blood [6]. However, the molecular signaling pathways involved in medulloblastoma metastasis are still largely unknown.

The Rho/Rho-associated coiled-coil containing protein kinase (ROCK) signaling pathway is best known through its action on the actin cytoskeleton. It is a key regulator in many biological processes, including cell adhesion, migration, and differentiation [7,8]. Active Rho GTPases lead to activation of the serine/threonine kinases ROCK1 and ROCK2, which phosphorylate several downstream substrates, including myosin light chain (MLC), myosin phosphatase target subunit (MYPT) and LIM domain kinases-1/2 [9,10]. The pathway is critical during embryonal development and is an important signal transduction component within the nervous system. ROCK1 and ROCK2 are ubiquitously expressed in most tissues, but ROCK2 is most highly expressed in muscle tissues and in the brain, which suggests that the protein might have a specialized role in these cell types [10]. ROCKs are dysregulated in a variety of cancer types and have been implicated in tumor motility, invasion, and growth [9,11]. ROCK signaling has also been associated with stiffening of the extracellular matrix, which can contribute to increased cell growth and more aggressive tumor behavior [12,13]. Preclinical studies have demonstrated a therapeutic potential of ROCK inhibition on tumor cell growth, migration and metastasis in a variety of cancers including brain tumors [14,15,16,17,18,19].

Here, we investigate the serine/threonine kinase ROCK as a therapeutic target in medulloblastoma. We demonstrate that inhibition of ROCK results in reduced medulloblastoma growth, especially for cells derived from Group 3 and 4 medulloblastoma. Additionally, we show that the ROCK inhibitor RKI-1447 suppresses ROCK-dependent signaling, tumorigenicity, and invasion by affecting RHOA, RHOB, vimentin, and JUN in vitro and inhibits tumor growth in vivo.

## 2. Results

### 2.1. ROCKs Are Expressed in Medulloblastoma Patient Samples and Cell Lines

In order to explore the importance of Rho/ROCK signaling in medulloblastoma, we analyzed mRNA expression levels of the downstream Rho activating kinases ROCK1 and ROCK2 in two different cohorts of medulloblastoma tissue samples, and in fetal and adult cerebellum. The expression of ROCK1 and ROCK2 showed significant differences between different molecular subgroups of medulloblastoma. The highest expression of ROCK1 was found in the Shh group while mRNA levels of ROCK2 were highest in Group 4 medulloblastomas (Figure 1A,B). This was consistent between the two investigated cohorts. The expression of ROCK1 was significantly higher in all four medulloblastoma subgroups compared to in adult cerebellum, but when comparing with the fetal cerebellum, only the Shh group of tumors showed a difference in the expression levels (Figure 1A). For ROCK2 expression, the adult cerebellum showed higher mRNA expression than the medulloblastoma subgroups Wnt, Shh and Group 3, while no difference was detected in comparison to Group 4. No differences were observed between ROCK2 in fetal cerebellum and any of the medulloblastoma subgroups (Figure 1A). mRNA expression of ROCK1 or ROCK2 was not significantly associated with overall survival in either cohort. As Rho/ROCK signaling is involved in migration and metastasis we investigated the expression in non-metastatic versus (vs) metastatic samples. For ROCK2 the metastatic samples expressed significantly higher mRNA levels compared to the non-metastatic samples, while no difference was detected for ROCK1 (Figure 1C, Appendix A). Publicly available expression cohorts on cell lines showed mRNA expression of both ROCK1 and ROCK2 in all accessible medulloblastoma cell lines: UW228, D283, D458, D425, and DAOY (Appendix A). Western blot analysis for ROCK1 and ROCK2 in a panel of medulloblastoma cell lines showed evident levels of protein expression from at least one of the two ROCKs in all cell lines analyzed, as well as activation of the downstream target MLC, however very low in the D458 cell line (Figure 1D, Appendix A).

### 2.2. Medulloblastoma Cell Lines Are Sensitive to ROCK Inhibition

To investigate the effects of ROCK inhibition on medulloblastoma cell growth, we tested the viability of medulloblastoma cells after incubation for 72 h with three different ROCK inhibitors. We used a panel of nine medulloblastoma cell lines derived from different subgroups, one supratentorial primitive neuroectodermal tumor (sPNET) cell line and two non-tumorigenic fibroblast cell lines (Appendix A). The three compounds tested were RKI-1447, a specific inhibitor of ROCK1 and ROCK2 (IC_50_ in cell-free assays of 14.5 and 6.2 nM respectively) [16], HA1077, an inhibitor of ROCKs (IC_50_ value for ROCK2 in cell-free assays of 0.3–1.9 µM) but also other kinases at higher concentrations [24,25], and the multi-kinase AGC-inhibitor AT13148 that inhibits ROCKs, p70S6K, protein kinase A (IC_50_s in cell-free assay = 3–8 nM) and AKT1,2 and 3 (IC_50_s in cell-free assay = 38, 402, and 50 nM, respectively) [19]. All three inhibitors induced a concentration-dependent decrease in medulloblastoma cell viability after 72 h of treatment. The mean IC_50_ for all investigated cancer cell lines was 7.28 µM (range 1.08–23.6 µM) for RKI-1447, 5.35 µM (range 1.03–14.1 µM) for AT13148 and 74.4 µM (range 33.8–124 µM) for HA1077 (Figure 2A–D, Appendix A). The mean IC_50_ for RKI-1447 and AT13148 did not differ, but both RKI-1447 and AT13148 showed a significantly lower mean IC_50_ than HA1077 in the cancer cell lines. Comparison in individual cell lines showed that RKI-1447 and AT13148 both were superior as inhibitors of medulloblastoma growth compared to HA1077 in all tested cell lines. For the comparison of RKI-1447 and AT13148, AT13148 showed a significantly lower IC_50_ in two of the cell lines (D283 and DAOY). No differences were observed in the remaining cell lines (Figure 2A–D, Appendix A). In the non-tumorigenic fibroblast cell lines, MRC-5 and nHDF, the mean IC_50_ value for RKI-1447 was 14.7 µM, which was significantly higher than the mean IC_50_ in the medulloblastoma cell lines, 7.28 µM. For AT13148 the mean IC_50_ values were 8.04 µM vs. 5.35 µM and for HA1077 100 µM vs. 74.4 µM (Figure 2A–D, Appendix A).

To investigate the effect of ROCK inhibitors in metastatic medulloblastoma, we compared ROCK inhibitors to standard cytostatic drugs in two pairs of cell lines derived from primary tumor and metastasis at recurrence from the same patients, as well as in the patient-derived cell line MB-LU-181, from a primary tumor with the ability to form metastases when xenografted in mice [26]. In the pair of Group 4 medulloblastoma cells, CHLA-01-MED and CHLA-01R-MED, RKI-1447 was significantly more effective in inhibiting cell growth in the relapse/metastatic cells (CHLA-01R-MED) compared to primary (CHLA-01-MED). None of the tested standard cytostatic drugs showed the same profile (Figure 2E,F). However, the same pattern was not observed in the Group 3 cell line pair D425 and D458, while vincristine and temozolomide had lower IC_50_ values in the relapse/metastatic cell line (D458) compared to the primary (D425), RKI-1447 and HA1077 showed the opposite profile (Figure 2E). MB-LU-181 was the most resistant cell line to cisplatin, temozolomide, and etoposide treatment, but showed intermediate sensitivity to the ROCK inhibitors and vincristine (Figure 2E).

### 2.3. RKI-1447 Inhibits ROCK Phosphorylation of Downstream MLC2

RKI-1447 efficiently inhibited medulloblastoma cell growth and is shown to be a highly potent and selective ROCK Type 1 inhibitor that binds to the ATP binding site of ROCK at the hinge region and the Asp-Phe-Gly (DFG) motif [16]. Based on the effects on cell growth and selectivity compared to AT13148 and HA1077 [16,19,24,25] RKI-1447 was selected for further studies. To confirm the mechanism of action on ROCK activity, we investigated the main downstream phosphorylated target of ROCKs, MLC2. RKI-1447 inhibited the phosphorylation of MLC2 (Ser19) in all investigated cell lines while no effect was observed on total MLC2 (Figure 3A, Appendix A). To further investigate the involvement of the ROCK2 in the decreased medulloblastoma cell growth upon treatment with RKI-1447, we inhibited ROCK2 using small interfering RNA (siRNA) targeting ROCK2. siRNA-mediated down-regulation of ROCK2 expression significantly suppressed the cell viability of medulloblastoma cells (Figure 3B). Two different siRNA pools were used for a knockdown to minimize the risk of off-target effects from siRNA and the downregulation of ROCK2 was confirmed on protein level (Figure 3C). Given the role of ROCK/MLC2 in cytoskeleton remodeling, we examined possible changes in cell morphology. We observed cells with elongated cell morphology after treatment with RKI-1447 for 24 h and 72 h (Figure 3D, Appendix A).

### 2.4. Inhibition of ROCK Impairs the Clonogenic Capacity, Migration, and Invasion of Medulloblastoma Cells In Vitro

To assess the inhibitory effect on tumorgenicity and migration/invasion we used clonogenic assay, migration and invasion assays with adherent medulloblastoma cells. RKI-1447 efficiently repressed the tumorigenic capacity of DAOY and UW228-3 in a concentration-dependent manner (Figure 4A,B). To examine the effects of RKI-1447 on migration and invasion we performed experiments with wound healing assay as well as in transwell chambers. RKI-1447 inhibited medulloblastoma cell migration in confluent cell monolayers, determined by the wound healing assay (Figure 4C–E). To verify equal conditions for the treatment group, the difference in wound size at the start of the experiments was tested. There was no difference in wound size at time 0 h when comparing the groups (Appendix A). Moreover, RKI-1447 significantly reduced the invasive ability of DAOY cells through a matrigel matrix towards medium with chemoattractant, as compared to vehicle control in transwell chambers (Figure 4F, Appendix A).

### 2.5. ROCK Inhibition Reduces the Expression of Key Molecules Involved in Medulloblastoma Progression and Metastasis

To identify genes and pathways involved in ROCK-dependent inhibition of medulloblastoma growth, we performed RNA sequencing after treatment with RKI-1447 in D425 medulloblastoma Group 3 cells. D425 cells were chosen since these cells are described to have metastatic features [27]. When compared with vehicle controls, D425 cells treated with 1 µM RKI-1447 for 24 h, displayed 995 genes with differential expression, 661 genes had decreased expression levels, whereas 334 genes had increased expression levels upon treatment with RKI-1447 (Appendix A). Among those with most significantly decreased expression levels in response to ROCK inhibition were *RHOB*, *RHOA*, *VIM*, *JUNB*, *JUN,* and *JUND*. Gene set enrichment analysis (GSEA) of hallmark gene sets representing well-defined biological processes showed that twelve gene sets were significantly downregulated (FDR *q* value < 0.25) while no significantly upregulated gene sets were identified (Table 1). Among the top downregulated hallmark pathways were several gene sets important for tumor progression and metastasis such as signaling through tumor necrosis factor α (TNFα) via nuclear factor κβ (NFκβ,), transforming growth factor β (TGFβ), P53, phosphatidylinositol-3-kinase (PI3K)/Akt/mammalian target of rapamycin (mTOR), mitotic spindle, epithelial to mesenchymal transition (EMT) and hypoxia (Table 1). GSEA charts and heatmaps for differential gene expression (DGE) in TNFα signaling via NFκβ and EMT are illustrated as examples in Figure 5A,B. Also, western blot analysis confirmed RKI-1447-mediated downregulation of RHOA, RHOB, JUN and vimentin in D425 after RKI-1447 treatment (Figure 5C, Appendix A). Expression of ROCKs were investigated in parallel: ROCK1 expression was upregulated at late time points (24 h and 72 h), whereas ROCK2 expression showed no change (Figure 5C, Appendix A). Additionally, a similar trend for both JUN and vimentin was detected in Med8a cells (Figure 5D, Appendix A). GSEA analysis using Reactome gene sets showed significant downregulation of 29 terms (FDR *q* value < 0.25). For example, pathways/processes associated with cell migration and Rho protein signaling were downregulated (Appendix A).

### 2.6. RKI-1447 Suppresses Medulloblastoma Growth In Vivo

To further investigate the therapeutic effects of ROCK inhibition on medulloblastoma growth, RKI-1447 was evaluated in NMRI nu/nu mice carrying established subcutaneous D425 tumors. The mice were treated with 184 µmol/kg RKI-1447 (80 mg/kg RKI-1447 2HCl 2H_2_O), administered by daily intraperitoneal injection and compared with tumor-bearing control mice. Treatment started when the tumor size exceeded 200 mm^3^ and continued for ten days. The tumor growth was significantly suppressed in the treatment group compared to the control group. The mean tumor volume index (TVI) was lower in the RKI-1447 treated group vs. the control group already after one day of treatment and throughout the treatment period (Figure 6A). At the end of treatment, day 10, the mean TVI for RKI-1447 treated mice was 52% compared to control mice (Figure 6A). In addition, tumor weight at sacrifice from mice receiving RKI-1447 was significantly lower compared to control mice (Figure 6B). The average tumor weight was 0.54 g in the treatment group and 0.94 g in the control group (Figure 6B). Cell proliferation, measured by Ki-67 positive cells, was significantly lower in tumors from RKI-1447 treated mice compared to tumors from control mice, 53% vs. 67% (Figure 6C,D). Both treatment groups of mice gained weight during the study period (mean weight gain 0.91 g vs. 2.1 g for RKI-1447 treated and control mice, respectively). However, minor transient weight loss was recorded among the RKI-1447 treated mice (Appendix A).

## 3. Discussion

Metastatic spread is the cause of almost all medulloblastoma related death [3,4,5]. Medulloblastoma metastasis is characterized by leptomeningeal dissemination. However, the process remains poorly understood. This metastatic process includes acquiring the ability to invade the extracellular matrix and migrate out from the central tumor mass which requires an increased motility of the tumor cells. Hence, the tumor cells must undergo extensive remodeling of their cytoskeletal structure, an action that depends upon massive changes in gene expression that include activation of programs such as EMT [28]. As Rho GTPases and the serine/threonine kinases, ROCK1 and ROCK2 have been implicated in cytoskeletal regulation involved in cell motility, invasion and proliferation, all key steps in metastasis [8], we investigated the therapeutic potential of targeting the Rho/ROCK pathway in medulloblastoma, which to our knowledge is unexplored.

Our analysis of expression arrays showed that Group 4 medulloblastomas have the highest mRNA expression of ROCK2 among the subgroups (Figure 1A,B). In a study comparing the proteome and phosphoproteome landscape in medulloblastoma, Group 4 medulloblastomas were shown to express high levels of protein and phosphorylation of the guanine-nucleotide exchange factors (GEFs), as well as proteins signaling downstream of GEFs including ROCK2 [29]. In line with this, another large genome-wide DNA methylation and gene expression study (using the same patient cohort as in Figure 1B,C), showed that cell migration was an enriched pathway in medulloblastoma Group 4 [20]. Both Group 3 and Group 4 medulloblastomas are described to be highly metastatic [20]. We observed an increased expression of ROCK2 in metastatic compared to non-metastatic tumors samples (Figure 1C). This suggests a possible role for ROCK2 in medulloblastoma metastasis. We, therefore, investigated the effects of three different ROCK inhibitors and demonstrated that ROCK inhibition can efficiently repress medulloblastoma cell growth (Figure 2A–D, Appendix A).

Dose-response curves for RKI-1447 in the nine tested medulloblastoma cell lines gave IC_50_ values ranging from 1.08 to 23.6 µM, which is higher than the IC_50_ for ROCK inhibition in cell-free assays. This could be due to pharmacokinetic properties but can also indicate that off-target effects could be involved in the repression of cell growth, especially in the higher concentration used. To confirm ROCK2-specific effects on medulloblastoma growth we performed siRNA experiments, and indeed observed a significant decrease in cell viability after ROCK2 downregulation (Figure 3B,C, Appendix A). Comparing previously studies using RKI-1447 in cancer, reported IC_50_ values from cell viability assay (WST-1, MTT) and clonogenic assay are comparable to or higher than what we observed in medulloblastoma cell lines [16,30].

An intriguing observation was the sensitivity to the ROCK inhibitors in the cell line pair isolated from a disseminated medulloblastoma belonging to Group 4: CHLA-01-MED (from original tumor) and CHLA-01R-MED (from malignant pleural effusion at the second recurrence that occurred despite intensive chemotherapy and irradiation [31]). CHLA-01R-MED showed a two-fold increase in sensitivity measured by IC_50_ for the highly specific ROCK inhibitor RKI-1447 compared to CHLA-01-MED. A similar pattern was shown for the other ROCK inhibitors tested (Figure 2E,F). In general, CHLA-01R-MED demonstrates increased resistance to cytotoxic drugs [31], which was confirmed by us (Figure 2E,F). In a study investigating altered proteins in metastatic medulloblastomas, the two CHLA-01-MED cell lines were compared. Genes related to focal adhesion and regulation of actin cytoskeleton as well as EMT (e.g., vimentin) were significantly enriched in CHLA-01R-MED compared to CHLA-01-MED [32]. Genes in these pathways/processes were indeed significantly downregulated in our DGE analysis on medulloblastoma cells treated with RKI-1447 (Figure 5, Table 1, Appendix A). We did not observe the same increased sensitivity in the cell line pair D425 and D458, however, their usefulness as models for primary/metastasis have been questioned since the primary (D425) is *TP53* mutated while the one derived from the metastasis (D458) is *TP53* wildtype [33]. The differences in *TP53* is a probable explanation for the increased sensitivity to temozolomide (Figure 2E). Moreover, also the primary cell line D425 has been described as metastatic [27].

Our results showed that RKI-1447 suppressed migration and invasion of adherent medulloblastoma cells (Figure 4). We also observed that RKI-1447 treated medulloblastoma cells displayed an elongated morphology, suggesting differentiation (Figure 3D, Appendix A). Tumor cells transmigrate in two different modes, either in a rounded bleb-based way which is Rho/ROCK dependent or in an elongated, protrusive mode which is independent of Rho/ROCK and relies on Rac [34]. Therefore, our findings may indicate that the elongated, protrusive mode of invasion is not inhibited by RKI-1447, which could explain why the cell migration/invasion is not blocked completely.

The GSEA showed that ROCK blockade by RKI-1447 treatment induced downregulation of several signaling pathways important for tumor progression and metastasis (Table 1). We functionally confirmed anti-migrative/invasive properties in vitro and inhibition of tumor growth in vitro and in vivo. This is in line with what other studies have reported in other cancer types [8,14,16,17]. We observed a significant downregulation of genes upstream from ROCK, *RHOA,* and *RHOB* after RKI-1447 incubation, which was confirmed on protein level (Appendix A, Figure 5). Conversely to RHOA and RHOC, RHOB has been described to be downregulated in several malignancies where its expression promotes apoptosis in cancer cell lines and inhibits invasion [35]. RHOB has therefore been suggested to have a tumor suppressor role, while RHOA and RHOC are considered oncogenes [36,37]. However, new studies are challenging these findings, suggesting that RHOB is an oncogene since studies have shown that RHOB has a tumor-promoting role in glioblastoma [38,39]. Paradoxical observations are reported for brain tumors [38,40] as well as for other cancer types, such as lung cancer, melanoma and breast cancer [36]. These contradictory observations suggest that RHOB may have different functions in cancer that are context-dependent and cell-type specific, possibly responding to signals in the tumor microenvironment and that RHOB can function both as an oncogene and a tumor suppressor in cancer [41]. We also observed a strong RKI-1447 mediated downregulation of the Jun transcription factor family *JUN*, *JUNB* and *JUND*, all components of the transcription factor activator protein-1 (AP-1). This finding suggests crosstalk between ROCK and c-Jun N-terminal kinases (JNK), which mainly act downstream of Rac, in parallel to the Rho/ROCK pathway [42]. This interaction may be partly responsible for the effects seen on proliferation since JNKs are important regulators of the cell cycle and are target genes of AP-1 [43]. The EMT marker vimentin was also inhibited after RKI-1447 treatment (Appendix A, Figure 5). Vimentin has previously been reported to be a target for ROCK phosphorylation. Phosphorylation of vimentin-Ser71 was observed in various cell types, such as glioma cells, suggesting that ROCK may have a role in the regulation of vimentin [10,44].

Finally, we also demonstrated that ROCK inhibition by RKI-1447 suppressed tumor growth in vivo. Treatment with RKI-1447 significantly inhibited the proliferation of established medulloblastoma xenograft tumors from the Group 3 cell line D425 (Figure 6). We observed a transient minor weight loss among the treated mice, an effect that could be attributed to the ROCK inhibition and the described vasodilatory effect [45,46]. One advantage of RKI-1447 over other ROCK inhibitors is that it targets both ROCK1 and ROCK2. Knockout studies of ROCK1 and ROCK2 have shown that ROCKs are essential for cell cycle progression and tumorigenesis, but in order to achieve efficient tumor growth inhibition, it is preferable to target both ROCK1 and ROCK2 to avoid redundancy [47]. However, targeting both ROCKs may also cause additional side effects [48]. ROCK inhibitors are currently in clinical use, mainly HA1077 (Fasudil) due to its vasodilatory properties in the treatment of cerebral vasospasm. Fasudil is described to be safe and well-tolerated with only mild side effects. However, Fasudil is usually prescribed for a short time span (two weeks). The safety and efficacy profile in long-term use remains to be evaluated [45,46]. Other ROCK inhibitors are in clinical trials for different indications. To the best of our knowledge, AGC kinase inhibitor AT13148 [19], included in our study, is the only ROCK inhibitor tested for cancer. A phase I study for advanced solid tumors has recently finished but no results have been reported so far (clinicaltrials.gov, identifier: NCT01585701).

Taken together, our results suggest that small molecules targeting the Rho/ROCK pathway represent a potential new treatment option for medulloblastoma patients. However, Rho/ROCK signaling is context-dependent and the usage of cell lines as model systems has limitations with regard to the absence of interaction with tumor microenvironment and the extracellular matrix. Additional studies are needed to further substantiate these results for example by using more clinically relevant low passage patient-derived xenograft (PDX) cells as well as in vivo models with a focus on metastatic tumors and later in clinical trials.

## 4. Materials and Methods

### 4.1. Cell Culture and Reagents

Nine medulloblastoma cell lines, derived from different subgroups (Shh, Group 3, and Group 4), with different genetic characteristics, and one sPNET cell line (PFSK-1) were used in the study. All cell lines, along with growth patterns, group classification and other characteristics, are listed in Appendix A. The cells were purchased from ATCC (ATCC-LGC Standards, Middlesex, UK), except for D425, D458, Med8a, and UW228-3, which were kindly provided by Dr. M. Nistér (Karolinska Institutet), MB-LU-181 that was established by us [26] and the normal human dermal fibroblast cell line (nHDF) that was purchased from PromoCell (Heidelberg, Germany). The cells were grown and maintained as follows: Med8a in Dulbecco’s modified Eagle’s medium (DMEM), DAOY and D283 in Minimum Essential Media (MEM), D425 and D458 in Richter’s improved MEM with zinc/DMEM, UW228-3 in DMEM/F12 and nHDF, MRC-5, and PFSK-1 in RPMI 1640, and supplemented with 10% (or 15% for Med8a, D425, D458) heat-inactivated fetal bovine serum (FBS), 2 mM L-glutamine, 100 IU/mL penicillin G, and 100 μg/mL streptomycin (all from Life Technologies Inc, Thermo Fisher Scientific, Stockholm, Sweden). CHLA-01-MED and CHLA-01R-MED spheres were cultured in DMEM/F12 with 20 ng/mL human recombinant epidermal growth factor (EGF, Chemicon, Merck Millipore, Solna, Sweden), 20 ng/mL human recombinant basic fibroblast growth factor (bFGF) (Gibco Life Technologies Inc), and B-27 Supplement (Gibco, Life Technologies Inc), in addition to L-glutamine, streptomycin and penicillin supplemented as described above. MB-LU-181 spheres were grown in Ultra-Low attachment 6-well plates (Corning, VWR, Spånga, Sweden) in UltraCULTURE™ cell culture medium (Lonza BioWhittaker Inc., VWR) supplemented with 20  ng/mL EGF, plus L-glutamine, streptomycin and penicillin as above. All cells were grown at 37 °C in a humidified 5% CO_2_ atmosphere. All media were purchased from Gibco (Life Technologies Inc). The identities of the cell lines were verified by short tandem repeat genetic profiling using the AmpFlSTR^®^ IdentifilerTM PCR Amplification Kit 2015 (Applied Biosystems, Thermo Fisher Scientific) and were routinely tested for mycoplasma (Mycoplasmacheck, Eurofin Genomics, Ebersberg, Germany). All experiments were executed in Opti-MEM supplemented with glutamine, streptomycin, and penicillin except for MB-LU-181 cells which were seeded in their own growing medium.

For in vitro use, RKI-1447 and AT13148 were purchased from Cayman Chemical (Ann Arbor, MI, USA), HA1077 dihydrochloride, cisplatin, and vincristine from LC Laboratories (Woburn, MA, USA) and temozolomide from Sigma-Aldrich (Solna, Sweden). RKI-1447, AT13148, temozolomide, and vincristine were dissolved in DMSO (Sigma-Aldrich), HA1077 in sodium chloride, and cisplatin in sterile water, further dilutions were done in Opti-MEM or PBS. The DMSO concentration was <1% *v/v* in all experiments. For the in vivo studies, RKI-1447 dihydrochloride (2H_2_O) was purchased from Tocris (R&D System, Minneapolis, MN, USA) and dissolved in sterile water with 0.1% Tween80 (Sigma-Aldrich).

For the siRNA transfections, cells were seeded in six-well dishes, left to attach, and transfected using Lipofectamine 2000 (Cell Signaling Technology, Leiden, The Netherlands) with 100 pmol predesigned siRNAs targeting human ROCK2 (siRNA (1), #5270301, Invitrogen, Life Technologies Inc), ROCK2 (siRNA (2), locus ID 9475, SR306287, OriGene Technologies, Rockville, USA) or non-targeting siRNA (6568s, Invitrogen, Life Technologies Inc) as control. After 72 h, cells were harvested and subjected to further analyses.

### 4.2. Cell Survival Analysis

Cell viability was assessed as a relative metabolic activity using the formazan-based assay WST-1 (Roche, Sigma-Aldrich). Cells were seeded into 96-well plates (5000–30,000 cells/well), adherent cells were left to attach, and then treated with increasing drug concentrations and incubated for 72 h. Analyses were done according to the manufacturer’s protocol and absorbance was measured at 450 and 650 nM using a VersaMax reader (Molecular Devices, San Jose, CA, USA). Cell viability is presented as % of untreated control cells, with blank values subtracted.

To determine colony formation 150–200 cells/well were seeded in 6-well plates (Cell+, Sarstedt, Solna, Sweden) and allowed to attach 6 h before drug or vehicle exposure for 72 h. After an additional 5–7 days of incubation in drug-free complete medium, cells were washed, fixed, stained with Giemsa (Sigma-Aldrich) and colonies (>75 cells) with at least 50% plate efficiency were counted.

### 4.3. Western Blot

Protein extracts were lysed on ice with Pierce RIPA buffer (Thermo Fisher Scientific) including Halt Protease and Phosphatase Inhibitor Cocktail (100X) (Thermo Fisher Scientific). Protein concentrations were analyzed using DC-Protein Assay (BioRad) following the manufacturer’s protocol. Between 15 µg to 30 µg of protein extracts (depending on the amount left-over and the concentration of the protein extracts) were diluted (1:3) in Blue Loading Buffer pack containing DTT according to the manufacturer’s protocol (New England Biolabs, Ipswich, MA, USA). Diluted samples were incubated for 10 min at 95 °C, loaded on a MiniProtean TGX 10% gel (BioRad Laboratories, AB, Solna, Sweden), and finally transferred to a Nitrocellulose membrane (GE Healthcare, Danderyd, Sweden) overnight at 22V. The membranes were blocked using non-fat dry milk of 5% in 0.1% TBST and incubated with primary antibodies against antibodies specified in Appendix A, overnight at 4 °C. HRP-conjugated anti-mouse or anti-rabbit (all from Cell Signaling Technology) diluted in blocking buffer were used as secondary antibodies. Membranes were incubated with ECL detection reagent SuperSignal West Pico Plus (Thermo Fisher Scientific) and subsequently, the membrane was exposed to a chemiluminescent machine (GE Healthcare). MagicMark XP (Thermo Fisher Scientific) was used to visualize protein sizes. Quantification of blots was done with densitometry measurements in ImageJ [49,50]. All full blots corresponding to the displayed western blot analysis are available in Appendix A.

### 4.4. Morphological Studies

Medulloblastoma cells were seeded in 25 cm^2^ flasks (100,000 cells), left to attach, and treated with RKI-1447 (IC_50_ from the viability assay 72 h and half of the IC_50_ value), and images were acquired with a phase-contrast microscope Nikon Eclipse Ts2, 20× objective (Nikon Instruments Inc., Melville, NY, USA).

### 4.5. Migration and Invasion Analyses

Wound assay was used to evaluate the migration capacity after RKI-1447 treatment in medulloblastoma cells. UW228-3 and DAOY cells were seeded in 6-well plates (500,000 cells/well) in complete media and left to attach for 24 h. A scratch was made using a 200 µL (UW228-3) or 1000 µL (DAOY) pipette tip, followed by drug treatment. The media was changed to media with 1% FBS to minimize proliferation. Images were obtained at the start and 18 h after the treatment using a phase-contrast microscope (Nikon Eclipse Ts2, at 10× for UW228-3 and at 4× for DAOY), and used for quantitative assessment of migrated area [51]. The assessment was done using automated analysis in TScratch software [52]. Three to four independent experiments were performed with four images per treatment.

Cell invasion assay was performed in 16-well CIM plates with transwell chambers and a microporous polyethylene terephthalate (PET) membrane, with pores of 8 µm in diameter coated with 30 µL 15% (equivalent to 1.2 mg/mL) matrigel (growth factor reduced, Corning) using an xCELLigence DP instrument for real-time analysis (ACEA Biosciences, Inc., San Diego, CA, USA). DAOY cells were starved in serum-free RPMI medium for 6 h, including a pre-treatment for 3 h with RKI-1447 or vehicle, seeded in the upper chamber in serum-free medium, 30,000 cells per well. The lower chamber contained complete RPMI medium (10% FCS) that had been used in the cultivation of nHDF, centrifuged to remove any remaining nHDF cells, the supernatant was frozen until the invasion experiment. The negative control contained cells but no chemoattractant in the lower chamber. Cell invasion was monitored every 2 h for a period of 40 h. All treatments were performed in at least triplicate and repeated three times.

### 4.6. RNA Sequencing, Differential Expression Analysis, and Gene Set Enrichment Analysis

Medulloblastoma cells D425 were treated with RKI-1447 1 µM or vehicle (0.01% DMSO) for 24 h, in triplicates. RNA was purified with RNeasy Kit (Qiagen, Sollentuna, Sweden). Total RNA was subjected to quality control with Agilent Tapestation according to the manufacturer’s instructions. Sequencing libraries were constructed using the TruSeq Stranded mRNA sample preparation protocol (including mRNA isolation, cDNA synthesis, ligation of adapters and amplification of indexed libraries). The yield and quality of the amplified libraries were analyzed using Qubit by Thermo Fisher Scientific and the Agilent Tapestation. Indexed cDNA libraries were normalized and combined and the pools were sequenced on the Illumina NextSeq 500 platform for a 75-cycle v2 sequencing run, generating 75 bp single-end reads. Basecalling and demultiplexing were performed using CASAVA software with default settings generating Fastq files for further downstream mapping and analysis.

Raw sequencing fastq files were aligned to the GRCh38 human reference genome using the STAR 2-pass approach [53]. Aligned reads were quantified using htseq-count [54] and differential gene expression was performed using DEseq2 [55]. After DESeq2 analysis, genes were ranked by adjusted *p* value and the sign of the log fold change and gene set enrichment analysis was performed using GSEA [56,57] using the Hallmark and Reactome gene sets from MSigDB [58].

### 4.7. Gene Expression Profiling

Gene expression profiling on medulloblastoma samples, presented in Figure 1A, were performed on Affymetrix 133plus2.0 arrays, MAS5.0 normalized, as described previously [22], samples are also used in [21,22,23]. Gene expression analysis on medulloblastoma samples presented in Figure 1B,C are from the analysis of publicly available gene expression data sets, acquired using the R2 microarray analysis and visualization platform (http://r2.amc.nl) with the Cavalli data set [20].

### 4.8. In Vivo Xenograft Studies

Immunodeficient nude mice (female 5–6 weeks old NMRI nu/nu, Scanbur, Sollentuna, Sweden) were subcutaneously injected with 5 × 10^6^ D425 cells in serum-free medium mixed at a ratio of 1:1 with Matrigel Basement Membrane Matrix High Concentration (Corning) at a total injection volume of 100 µL, into the left flank under anesthesia with isoflurane and medical air. Once the estimated tumor volumes exceeded 200 mm^3^, the mice were randomized to the treatment group (*n* = 11) or the control group (*n* = 10). The mean start volume was 240 mm^3^. RKI-1447 treatment was given as intraperitoneal injections daily for ten consecutive days. The duration of treatment was restricted by tumor size in the control group. Tumors were measured by caliper daily, and tumor volume was calculated using the formula: (width)^2^ × length × 0.44. The animals were monitored for signs of toxicity including weight loss. On day 10 after treatment started, the animals were sacrificed. Tumors were weighed and dissected in smaller parts, and either frozen or fixed in formaldehyde. Tumor volume index (TVI) was calculated as the measured tumor volume each day relative to the volume on day 0 when the treatment started. Mice that had not developed any tumor within one month from the injection of cells were excluded from the study.

Animals were maintained at a maximum of six per cage and given sterile water and food ad libitum. The animal experiment was approved by the Stockholm ethics committee for animal research (no. N231/14), appointed and under the control of the Swedish Board of Agriculture and the Swedish Court. The animal experiments presented herein were in accordance with national regulations (SFS no. 2018:1192 and SFS no. 2019:66).

### 4.9. Immunohistochemistry

Formalin-fixed and paraffin-embedded sections from D425 xenografts were deparaffinized and rehydrated before antigen retrieval in citrate buffer (pH 6) (Sigma). Endogenous peroxidase activity was blocked using 0.5% H_2_O_2_, and biotin was blocked using 1% Bovine Serum Albumin in Tris-buffered saline (TBS) 30 min. Tumor sections were incubated overnight at 4 °C with primary antibody against Ki-67 (Dako, Agilent Technologies, Inc., Santa Clara, CA, USA, 1:800, Appendix A). Sections were then incubated for 30 min at room temperature with a biotin-conjugated secondary antibody (anti-rabbit 1:200, Vector Laboratories). After incubation with ABC complex (Vector Laboratories, Burlingame, CA, USA), the sections were developed for 3 min using diaminobenzidine (DAB Peroxidase Substrate Kit, Vector Laboratories) and then counterstained with Mayer’s hematoxylin (Histolab). Quantification was done by counting the number of positively stained cells and the total number of tumor cells in four hotspot fields per slide using Olympus BH-2 (Olympus, Solna, Sweden), at 40× magnification. Results are provided as the proportion of positively stained cells.

### 4.10. Statistical Analysis

The IC_50_ values (inhibitory concentration 50%) were calculated from log concentration-effect curves in GraphPad Prism (GraphPad Software, San Diego, CA, USA) using non-linear regression analysis. Comparison between two groups was done with unpaired *t*-Test, for multiple *t*-Tests the Holm-Sidak method was used to correct for multiple comparisons and for groups with unequal variation Welsh correction. For three or more groups one-way ANOVA, or alternately two-way ANOVA for treatment groups with repeated measurements over time, followed by Bonferroni multiple comparisons post-Test was used. All tests were two-sided and carried out in GraphPad Prism.

## 5. Conclusions

Our results demonstrate that ROCK inhibition represses medulloblastoma proliferation and invasion. The findings show that pharmacological ROCK inhibition can be a potential therapeutic approach that targets not only tumor growth but also inhibits the invading capacity of medulloblastoma cells. More studies are needed to validate these results.

## Figures and Tables

**Figure 1 cancers-12-00073-f001:**
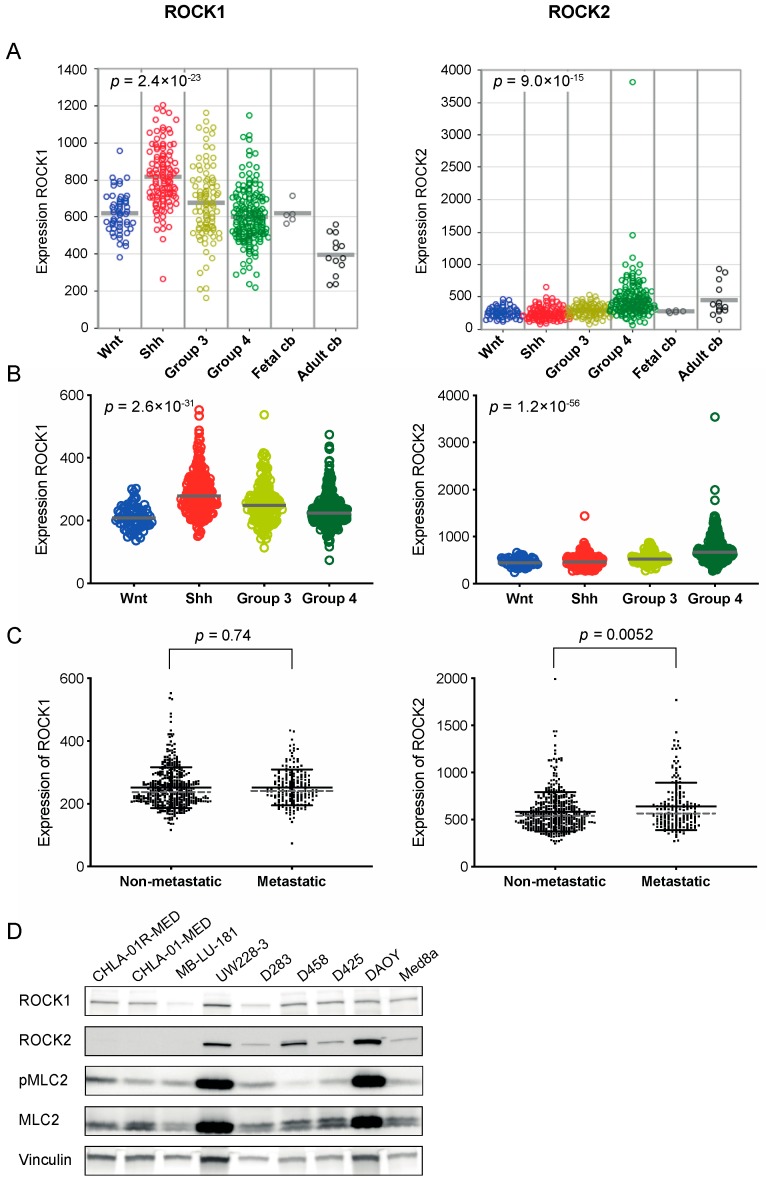
ROCK1 and ROCK2 are expressed in medulloblastoma tumors. (**A**,**B**) mRNA expression of ROCK1 and ROCK2 in medulloblastoma molecular subgroups (Wnt, Shh, Group 3 and Group 4) in two different patient cohorts and in fetal and adult cerebellum (cb) [20,21,22,23]. For the cohort shown in A, *n* = 423 primary medulloblastomas including Wnt *n* = 53, Shh *n* = 112, Group 3 *n* = 94 and Group 4 *n* = 164, plus fetal cb (*n* = 5) and adult cb (*n* = 13), and in B *n* = 763 primary medulloblastomas including Wnt *n* = 70, Shh *n* = 223, Group 3 *n* = 144 and Group 4 *n* = 326. P values from one-way ANOVA across the four medulloblastoma subgroups. Comparing ROCK1 expression in fetal cb tissue with medulloblastoma tumor samples showed no significant differences except when compared to Shh medulloblastomas (*p* = 0.0082). Moreover, all medulloblastoma subgroups displayed higher expression of ROCK1 than the adult cb (adult cb vs. all individual subgroup *p* < 0.001). For ROCK2 expression no differences were detected between the medulloblastoma subgroups and fetal cb, however, adult cb showed higher expression than the Wnt, Shh and Group 3 subgroups (adult cb vs. Wnt, Shh and Group 3, respectively *p* < 0.001). The center lines represent the data median (**A**,**B**). (**C**) mRNA expression of ROCK1 and ROCK2 in tumor samples from non-metastatic tumors (*n* = 397) and metastatic tumors (*n* = 176) [20]. ROCK2 expression was significantly higher in metastatic compared to non-metastatic samples, assessed with a *t*-Test. Both mean and SD (represented by full black lines), as well as median (dotted grey line), are shown (mean values for ROCK2 expression in non-metastatic vs. metastatic samples were 583 vs. 640, and median values were 541 vs. 562 respectively, ROCK2 mRNA values are available in Appendix A). (**D**) Western blot analysis showing protein expression of ROCK1 (160 kDa), ROCK2 (160 kDa), phosphorylated MLC2 (Ser19) (18 kDa), total MLC2 (18 kDa) and loading control vinculin (124 kDa) in all tested medulloblastoma cell lines. For densitometric analyses of protein expression see Appendix A.

**Figure 2 cancers-12-00073-f002:**
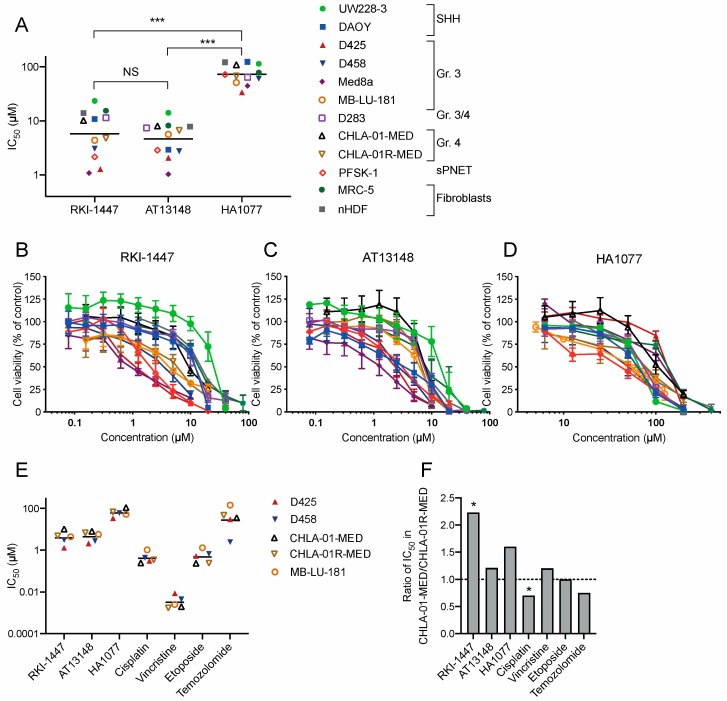
Pharmacological ROCK inhibition suppresses medulloblastoma growth. (**A**) IC_50_ (µM) for ROCK inhibitors RKI-1447, AT13148 and HA1077 in a panel of nine medulloblastoma cell lines, one sPNET cell line and two non-tumorigenic fibroblast cell lines after 72 h drug exposure. The mean IC_50_ values for RKI-1447 and AT13148 did not differ but both RKI-1447 and AT13148 showed lower mean IC_50_ values compared to HA1077 in cancer cell lines (one-way ANOVA *p* < 0.001, with Bonferroni posttest RKI-1447 vs. AT13148 *p* > 0.999, RKI-1447 vs. HA1077 *p* < 0.001 and AT13148 vs. HA1077 *p* < 0.001). Comparing individual cell lines showed that RKI-1447 and AT13148 were superior compared to HA1077 (one-way ANOVA with Bonferroni posttest *p* < 0.001). When comparing RKI-1447 and AT13148 in each cell line, AT13148 was more potent in inhibiting cell growth compared to RKI-1447 in two cell lines (DAOY and D283) (one-way ANOVA with Bonferroni posttest: DAOY: *p* = 0.0023, D283: *p* = 0.0088). RKI-1447 showed a significantly higher mean IC_50_ value in the non-tumorigenic fibroblast cell lines, MRC-5 and nHDF compared to the mean IC_50_ value in the medulloblastoma cell lines (*t*-Test, *p* = 0.017). (**B**–**D**) Dose-response curves for cell viability after 72 h for RKI-1447, AT13148 and HA1077 treatment in the same cell line panel (identically color-coded as in **A**). (**E**) IC_50_ (µM) for ROCK inhibitors RKI-1447, AT13148 and HA1077 and the standard cytotoxic drugs cisplatin, vincristine, etoposide and temozolomide in two pairs of cell lines from primary/metastatic samples: D425/D458 and CHLA-01-MED/CHLA-01R-MED, and one patient-derived cell line from a primary tumor but with metastatic features, MB-LU-181. (**F**) The ratio between IC_50_ values from CHLA-01-MED and CHLA-01R-MED. RKI-1447 showed a significantly lower IC_50_ in the metastatic cell line compared to the primary (*t*-Test *p* = 0.034) while cisplatin produced a significantly higher IC_50_ in the metastatic cell line compared to the primary (*t*-Test *p* = 0.022). (**A**–**F**) Cell viability was determined with the WST-1 assay. NS = non-significant, * *p* < 0.05 and *** *p* < 0.001. All concentrations were tested in at least duplicates and the experiments were repeated at least three times, in (**A**,**E**) the line represents the mean and in (**B**–**D**) mean with S.E.M. are displayed.

**Figure 3 cancers-12-00073-f003:**
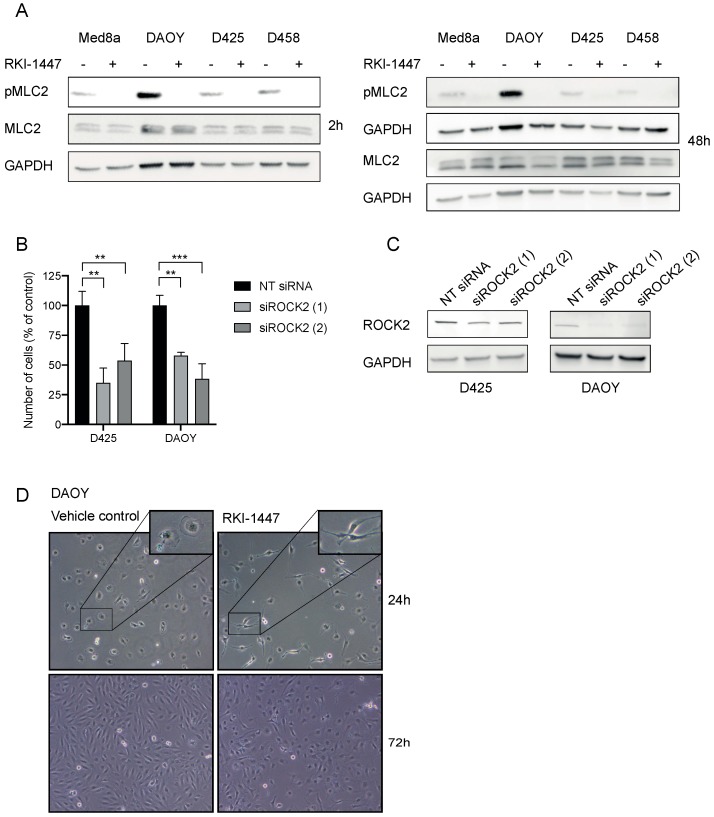
RKI-1447 inhibits phosphorylation of MLC2 in medulloblastoma cells (**A**) RKI-1447 inhibited downstream signaling in the Rho/ROCK pathway assessed by protein expression of phosphorylated MLC2 (Ser19) (18 kDa) compared to total MLC2 (18 kDa) and loading control GAPDH (36 kDa) at 2 h incubation, as well as phosphorylated MLC2 (Ser19) compared to total MLC2 (18 kDa) and GAPDH at 48 h treatment. Protein expression was investigated in RKI-1447 and vehicle-treated Med8a (1 µM), DAOY (10 µM), D425 (1 µM) and D458 (3 µM) with western blot. For quantification with densitometry see Appendix A. (**B**) siRNA-mediated down-regulation of ROCK2 expression suppressed medulloblastoma cell viability 72 h after transfection using two different siRNA compared with non-targeting (NT) control (one-way ANOVA with Bonferroni posttest, D425 *p* = 0.0022, NT control vs. siRNA1 *p* = 0.0015 and vs. siRNA2 *p* = 0.0086, DAOY *p* = 0.0004, NT control vs. siRNA1 *p* = 0.0022 and vs. siRNA2 *p* = 0.0003). Mean and SD of three experiments are displayed. ** *p* < 0.01 and *** *p* < 0.001. (**C**) The siRNA knockdowns of ROCK2 were confirmed with western blotting on ROCK2 (160 kDa) with GAPDH (37 kDa) as loading control at 72 h. Quantification with densitometry of protein expression of ROCK2 normalized to loading control showed for D425 NT control: 1 vs. siROCK2 (1) 0.44 and vs. siROCK2 (2) 0.52, for DAOY 1 vs. 0.11 and vs. 0.17 (Appendix A). (**D**) Morphological elongation of cells was evident in DAOY in response to RKI-1447 (10 μM) compared to vehicle-treated cells (24 h and 72 h). Images were acquired using a phase-contrast microscope (20× objective).

**Figure 4 cancers-12-00073-f004:**
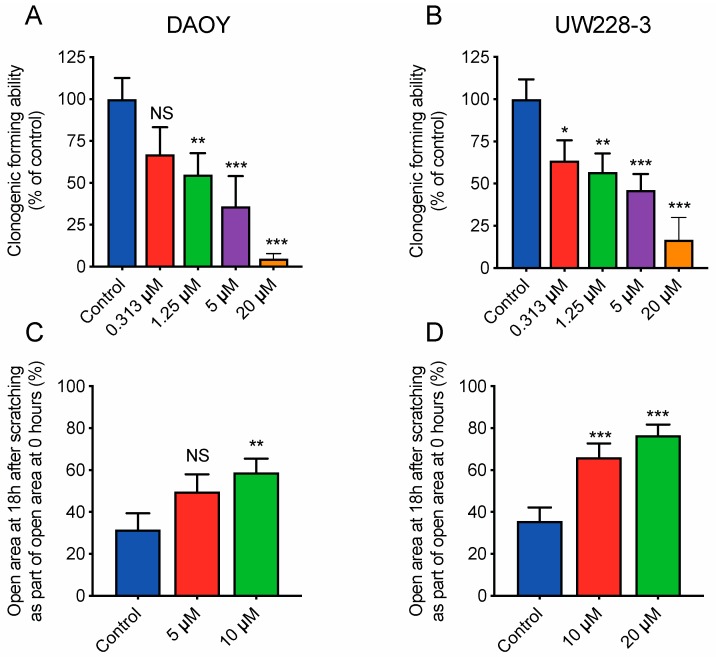
Inhibition of ROCK impairs the clonogenic capacity and migration/invasion of medulloblastoma cells (**A**,**B**) Tumorigenic capacity, shown as clonogenic forming ability, was inhibited in adherent medulloblastoma cell lines after 72 h of RKI-1447 exposure. RKI-1447 treatment in different concentrations were compared to vehicle control (one-way ANOVA with Bonferroni posttest *p* < 0.001 for both cell lines, DAOY: control vs. 0.31 µM *p* = 0.055, 1.25 µM *p* = 0.0089, 5 µM *p* < 0.001, 20 µM *p* < 0.001 and UW228-3: control vs. 0.31 µM *p* = 0.013, 1.25 µM *p* = 0.0043, 5 µM *p* < 0.001, 20 µM *p* < 0.001). IC_50_ for DAOY was 1.43 µM and 1.84 µM for UW228-3, respectively. The treatments were tested in at least triplicates and the experiments were repeated three times. Mean with S.D. are displayed. (**C**,**D**) RKI-1447 inhibited medulloblastoma migration shown as the relative area of migrated cells 18 h after scratching, as measured by wound assay (one-way ANOVA with Bonferroni posttest, DAOY: *p* = 0.012, control vs. 5 µM *p* = 0.053, control vs. 10 µM *p* = 0.009 and UW-228-3: *p* < 0.001, control vs. 10 µM and control vs. 20 µM *p* < 0.001). Four images were acquired per treatment and the experiment was repeated three-four times. Values presented are mean with S.D. (**E**) Example images from wound assay on DAOY cells at 0 and 18 h, with analysis from TScratch software, acquired at 4× objective. (**F**) RKI-1447 repressed the invasion ability in medulloblastoma cells. Real-time invasion analysis after treatment with 5 μM RKI-1447 in DAOY cells in transwell cell chambers coated with 15% matrigel, monitored in the xCELLigence RTCA system (two-way ANOVA with Bonferroni post-Test, *p* = 0.021 for treatment over the 40 h period, *p* < 0.05 from 23 h and forward for both control vs. 5 µM, *p* = 0.0005 40 h). The effect on invasion from 10 µM RKI-1447 are shown in Appendix A. Mean with S.E.M. of three independent experiments are shown. NS = non-significant, * *p* < 0.05, ** *p* < 0.01 and *** *p* < 0.001.

**Figure 5 cancers-12-00073-f005:**
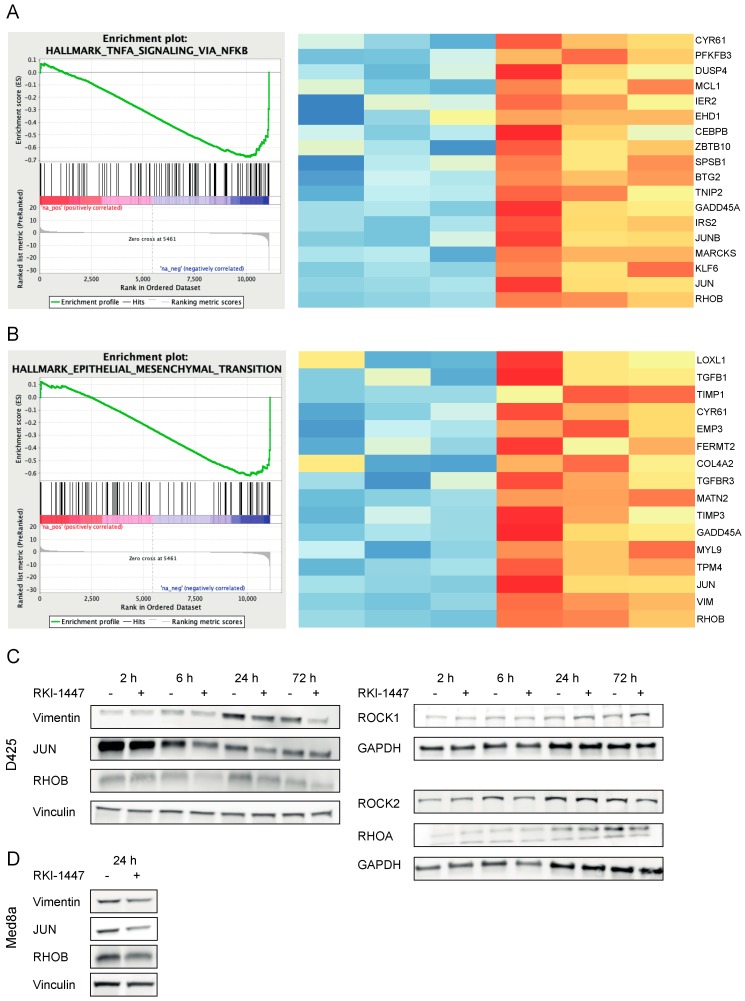
RKI-1447 downregulates TNFα via NFκβ and EMT signaling by inhibiting RHO, JUN, and vimentin. (**A**,**B**) GSEA of downregulated genes in RKI-1447 treated D425 cells (1 µM for 24 h) demonstrated over-representation of genes related to TNFα signaling via NFκβ (**A**) and EMT (**B**). Left panel: Enrichment plot from GSEA software, gene set systematic name (MSigDB): M5930 and M5890. Right panel: Heatmaps of core enriched genes from the same gene sets, color labeled by deviation from the average across all samples. Red color = above average, blue color = below average. mRNA expression was assessed with RNA Seq. (**C**,**D**) Western blot analysis on RHOA (21 kDa), RHOB (21 kDa), JUN (40–50 kDa), vimentin (53 kDa), ROCK1 (160 kDa), ROCK2 (160 kDa) and loading control GAPDH (26 kDa) or vinculin (124 kDa) protein expression in D425 cells treated with 1 µM RKI-1447 at different time points (indicated in the figure) (**C**) and RHOB (21 kDa), JUN (40–50 kDa), vimentin (53 kDa) and loading control vinculin (124 kDa) protein expression in Med8a cells treated with 1 µM RKI-1447 for 24 h (**D**). For densitometric analyses see Appendix A.

**Figure 6 cancers-12-00073-f006:**
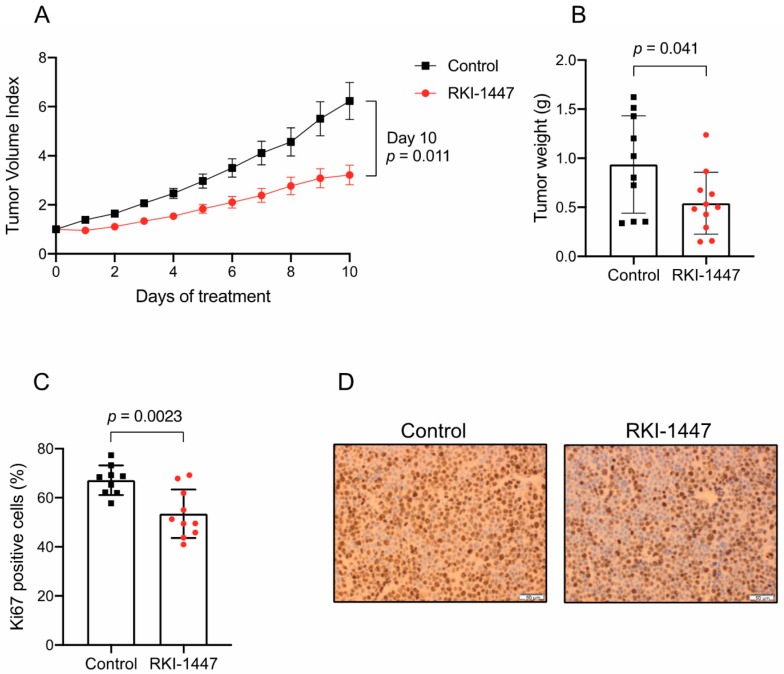
Pharmacological inhibition of ROCK reduces tumor growth in vivo. (**A**) RKI-1447 significantly impaired the growth of established human medulloblastoma xenografts in NMRI nu/nu mice (multiple *t*-Tests on tumor volume index (TVI) day 1 to 10 gave the following adjusted *p* values: day 1: *p* < 0.001, day 2: *p* = 0.0028, day 3: *p* = 0.0028, day 4: *p* = 0.0063, day 5: *p* = 0.015, day 6: *p* = 0.018, day 7: *p* = 0.018, day 8: *p* = 0.018, day 9 *p* = 0.018 and day 10: *p* = 0.011). Mean of TVI with S.E.M. are displayed. TVI was defined as the tumor volume at a given day relative to tumor volume on the day of inclusion. (**B**) Tumor weight at sacrifice (day 10) for RKI-1447 compared to control (*t*-Test *p* = 0.041). (**C**,**D**) Proliferative cells in tumors from control and RKI-1447 treated mice, measured by Ki67 immunohistochemistry (*t*-Test *p* = 0.0023). Means with S.D. are displayed in (**C**) and representative images (acquired at 40×) in (**D**). Mice were engrafted with 5 × 10^6^ D425 cells subcutaneously and randomized to a daily intraperitoneal injection of RKI-1447 dihydrochloride (2H_2_O) (80 mg/kg, *n* = 11) or control (*n* = 10), starting when the estimated tumor volume exceeded 200 mm^3^.

**Table 1 cancers-12-00073-t001:** Gene Set Enrichment Analysis (GSEA) results according to the MSigDB Hallmark gene sets. GSEA of downregulated genes in RKI-144 treated D425 cells (1 µM for 24 h) compared to vehicle control. NES = normalized enrichment score. Enrichments were considered significant if false discovery rate (FDR) < 0.25. Nom = nominal.

	Downregulated Hallmark Pathway	NES	FDR *q* Value	Nom *p* Value
1	TNF-α signaling via NFκβ	−1.68	0.062	0.003
2	Apoptosis	−1.64	0.056	0.004
3	TGF-β signaling	−1.62	0.045	0.004
4	IL2 STAT5 signaling	−1.57	0.069	0.007
5	P53 pathway	−1.54	0.078	0.006
6	PI3K Akt mTOR signaling	−1.53	0.074	0.013
7	Mitotic spindle	−1.52	0.068	0.003
8	Epithelial mesenchymal transition	−1.51	0.067	0.015
9	Oxidative phosphorylation	−1.51	0.060	0.008
10	Apical junction	−1.44	0.118	0.021
11	Complement	−1.36	0.213	0.057
12	Hypoxia	−1.35	0.214	0.049

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
