# Peer review of "Inhibition of Rho-Associated Kinase Suppresses Medulloblastoma Growth"

_cancers, 2019, doi:10.3390/cancers12010073_

Round 1

Reviewer 1 Report

Cecilia Dyberg et al in this work described the inhibitors of Rho associated kinases (ROCKs) in the medulloblastoma growth and found that one specific ROCK inhibitor RKI-1447 inhibited the  proliferation and metastasis in vitro and suppressed medulloblastoma growth and proliferation

 in vivo. Futhermore, they found that ROCK2 mRNA expression is upregulated in group 4  primary medulloblastoma and metastatic medulloblastoma. Their results could presents a new therapeutic treatment for the metastatic medulloblastoma. Some comments:

Fig1D, D458 with high ROCK1 and ROCK2, while decreased p-MLC2, which is not consistent with the description in the main text “ the two ROCKs in all cell lines analyzed, as well as activation  of the downstream target MLC”  and how this can be interpreted in the inhibitorPKI-1447 in D458 data in Fig2. Is there any known medulloblastoma data bank that the authors can extract the data to see whether ROCK1 and ROCK2 expression supporting the data in Fig 1A-D. Fig4F, Is there any difference between the PKI-1447 5um and 10uM? If not, there is no need to show both data here.

Author Response

Fig1D, D458 with high ROCK1 and ROCK2, while decreased p-MLC2, which is not consistent with the description in the main text “ the two ROCKs in all cell lines analyzed, as well as activation  of the downstream target MLC” and how this can be interpreted in the inhibitorPKI-1447 in D458 data in Fig2.

Response: We thank the Reviewer for the comment and acknowledge that D458 has very low phosphorylated MLC2, we have revised this in the text (page 3, line 98). The expression of ROCK itself does not necessarily lead to active downstream signaling, ROCK needs to be activated, most commonly by Rho-GTP, in order to phosphorylate downstream targets which do not only include MLC2 but also LIM kinases. So even if phosphorylated MLC2 is low in D458 the cells can still be dependent on the protein or another phosphorylation target of ROCK, which could explain the effects from RKI-1447, or alternatively the effect is off-target related, which is further discussed in the revised manuscript. 

Is there any known medulloblastoma data bank that the authors can extract the data to see whether ROCK1 and ROCK2 expression supporting the data in Fig 1A-D.

Response: We thank the Reviewer for this suggestion and have found mRNA expression for some of the medulloblastoma cell lines used in the study in the Broad Institute Cell Line Encyclopedia, available at www.depmap.com. These data are now described in the text (page 2, line 92) and is shown Figure S1A.

*Fig4F, Is there any difference between the PKI-1447 5um and 10uM? If not, there is no need to show both data here.

Response: There is no difference in the effect on invasion from RKI-1447 5 µM and 10 µM, we have moved the 10 µM dose to Figure S5 according to the Reviewers suggestion.

Reviewer 2 Report

The very well-written Dyberg et al. manuscript describes the study of the role of ROCKs in medulloblastoma using a pharmacological approach. As stated in their title, they conclude that the "inhibition of Rho-associated kinase suppresses medulloblastoma growth". This study and the conclusion of this study should be a great interest to the cancer community at large, and the medulloblastoma one in particular. There are some issues that would need to be discussed though, as described below.

The main issue relates to assessing whether the suppressive effect observed on medulloblastoma cell growth observed upon treating medulloblastoma cells with ROCK inhibitors is due either to the specific inhibition of ROCKs, or to a non-specific toxic effect associated with the use of these ROCK inhibitors at the concentrations (>5 uM) used by the authors in this study. Specifically:

1) In the entire manuscript, the authors do not discuss, let alone address, the possibility of a non-specific effect. The fact that RKI-1447 is a specific inhibitors in cell-free assays at concentration ranging from 14.5 to 6.2 nM cannot justify ignoring the possibility of a potential non-specific effect in cell assays when used at 10 uM. Similarly, demonstrating that the phosphorylation status of MLC2 is indeed modulated (Fig. 3a) does show that ROCK inhibition is indeed occurring at this high concentration but it does not demonstrate that the specific inhibition of ROCK is responsible for the cell growth effect. Comments are warranted.

"All three inhibitors induced a concentration-dependent decrease in medulloblastoma cell viability after 72 h of treatment."

Do the authors assume that all these observations for all three inhibitors in all medulloblastoma cell lines are due to specific ROCK inhibition? Or could some of these observations be due to a non-specific effect, e.g. the cell viability effect observed in UW228-3 cells upon 23.5 uM RKI-1447 treatment? Again, comments are warranted.

2) Dyberg et al. observed that "the mean IC50 for all investigated cell lines was 7.28 μM (range 1.08 – 23.5 μM) for RKI-1447".

How does their observations compare to RKI-1447 IC50 estimates in other cell lines and cell assays in other labs? A brief review of the literature suggests an IC50 value of ~0.7 uM in breast cancer cells (PMID: 22846914) or ~0.8 uM in clear cell renal cell carcinoma (PMID: 27841867).

Similar reviews of the literature and previously described IC50 (in cell assays) should be described for the other ROCK inhibitors to help the readers assess the extent to which the observations could be due to the specific inhibition of ROCKs, or to a non-specific toxic effect.

3) Several approaches could have been considered by the authors to address this issue empirically. For example:

-Are the inhibitory effects associated with ROCK inhibitors at these concentrations specifically observed in these medulloblastoma cell lines? Or are they actually observed in any cell line? Normal fibroblast, nHDF, is mentioned in Table S1 but is not shown in Fig 2. Were ROCK inhibitors tested on nHDF cells?

-The study is solely based on the use of a pharmacological approach. Does the genetic perturbation ROCK genes (KO or overexpression) results in the same observations?

-Comparing the IC50 observed for a ROCK-WT medulloblastoma cell line in comparison to its ROCK-KO counterpart could help demonstrate the specificity of the pharmacological effect.

-How about testing the FDA-approved, selective ROCK2 inhibitor KD025?  It would be interesting to assess whether KD025 has any effect in medulloblastoma cells where ROCK2 is expressed (DAOY, UW228-3), or not (CHLA-01, MB-LU-181).

Other comments:

4) In the in vivo experiment, which is limited to 10 days only, the authors observed a tumor growth suppressive effect but, in addition, they also observed a mild transient weight loss, suggesting a potential toxic effect.

Is the 10 day-limit of the experiment associated with avoiding a potentially more severe effect upon prolonged treatment?

Could this effect be attributed to the specific inhibition of ROCKs? Or could it be a non-specific toxic effect?

On the same note, the authors should consider discussing the genetic studies of ROCKs in mouse and cell and the phenotypes associated with the ROCK1/2 genetic KOs (including double KO), e.g., PMID:26765561 or PMID:30848941. In light of these observations, potential implications (side-effects?) for using ROCK inhibition as a therapeutic approach could be discussed.

5) Figure 1 and methods section 4.7 about the "Gene expression profiling": The authors should clarify, for each of the panel A and B,  the exact source of the data (Refs 20 only, or all four Refs 20-23) as well as the exact methods used for analysis (refering to all four Refs 20-23 is too vague), so that the readership has all the info to recapitulate the experiment.

6) The authors state "ROCK2 expression was significantly higher in metastatic compared to non-metastatic samples, assessed with t-test."

A P = 0.005 on Fig. 1C is reported, demonstrating the statement, but a visual inspection of Fig. 1C does not show an obvious difference. The median values actually appear to be very similar. This visual impression may be due to the fact that only outliers data points are shown, not the bulk of them. Could the authors show all the data points maybe? In addition to the median, could the authors report the mean? Can the authors share the raw data in an excel file (normalized Affymetrix values for the non-metastatic tumors (n = 397) and metastatic tumors (n = 176)) as part of the review process. They may actually consider adding this excel file as Sup File in the manuscript?

7) Fig 4E: At the beginning of the experiment (0 h), the wound appears to be significantly bigger in the RKI condition. At 18 hours, the size of the wound appears to have been reduced essentially by ~the same length in both conditions. If the size of the wound at 0h is bigger in the RKI condition than in the control condition, then the % of area covered after 18 hours would be lower in the control condition, even if the cells were actually migrating at the same rate. Using the % of open area as a reference point is only valid if the size of the wound is similar and reproducible in all the test and control experiments.

"Four images were acquired per treatment and the experiment was repeated three-four times." For each of the 12 (or 16?) images in each of the conditions tested, what were the estimates for the surface areas (not the %) of the wound measured at 0h?

Minor comments:

8) The authors should comment on the fact that these established cells lines are not ideal representatives of primary medulloblastoma tumors, as illustrated by clustering analysis of their genome/expression profiles in modern next-gene sequencing studies (they do not cluster with their respective medulloblastoma subgroup primary tumors). Some in the field would even argue that studies in these cells lines are not "translatable" to primary medulloblastoma biology. Hence, as a complement to their final remark ("More studies are needed to validate these results."), the authors could comment on the opportunity to test ROCK inhibitors on low-passage PDX lines.

9) Fig 3a: total MLC at 48 hours could have been shown as well.

Author Response

The very well-written Dyberg et al. manuscript describes the study of the role of ROCKs in medulloblastoma using a pharmacological approach. As stated in their title, they conclude that the "inhibition of Rho-associated kinase suppresses medulloblastoma growth". This study and the conclusion of this study should be a great interest to the cancer community at large, and the medulloblastoma one in particular. There are some issues that would need to be discussed though, as described below.

The main issue relates to assessing whether the suppressive effect observed on medulloblastoma cell growth observed upon treating medulloblastoma cells with ROCK inhibitors is due either to the specific inhibition of ROCKs, or to a non-specific toxic effect associated with the use of these ROCK inhibitors at the concentrations (>5 uM) used by the authors in this study.

Specifically:

1) In the entire manuscript, the authors do not discuss, let alone address, the possibility of a non-specific effect. The fact that RKI-1447 is a specific inhibitors in cell-free assays at concentration ranging from 14.5 to 6.2 nM cannot justify ignoring the possibility of a potential non-specific effect in cell assays when used at 10 uM. Similarly, demonstrating that the phosphorylation status of MLC2 is indeed modulated (Fig. 3a) does show that ROCK inhibition is indeed occurring at this high concentration but it does not demonstrate that the specific inhibition of ROCK is responsible for the cell growth effect. Comments are warranted.

"All three inhibitors induced a concentration-dependent decrease in medulloblastoma cell viability after 72 h of treatment."

Do the authors assume that all these observations for all three inhibitors in all medulloblastoma cell lines are due to specific ROCK inhibition? Or could some of these observations be due to a non-specific effect, e.g. the cell viability effect observed in UW228-3 cells upon 23.5 uM RKI-1447 treatment? Again, comments are warranted.

Response: We thank the Reviewer for bringing up this relevant concern, we have added text in the discussion regarding this issue (page 12, line 383-). We acknowledge that there can be off-targets effect from the inhibitors used in the study, especially in the higher concentrations. To address this issue we also have performed new experiments, suggested by the Reviewer, see below.

2) Dyberg et al. observed that "the mean IC50 for all investigated cell lines was 7.28 μM (range 1.08 – 23.5 μM) for RKI-1447".

How does their observations compare to RKI-1447 IC50 estimates in other cell lines and cell assays in other labs? A brief review of the literature suggests an IC50 value of ~0.7 uM in breast cancer cells (PMID: 22846914) or ~0.8 uM in clear cell renal cell carcinoma (PMID: 27841867).

Similar reviews of the literature and previously described IC50 (in cell assays) should be described for the other ROCK inhibitors to help the readers assess the extent to which the observations could be due to the specific inhibition of ROCKs, or to a non-specific toxic effect.

Response: We thank the Reviewer for this important comment and we have added the suggested comparisons with previously published studies of RKI-1447 in other cancer types to the discussion (page 12, line 389-). The reported IC50 for the breast cancer cell line MDA-MB-231 is 0.7 µM (PMID: 22846914) and 0.8 – 3.5 µM in the renal cell carcinoma cell lines RCC10, RCC4 and 786-O (PMID: 27841867). However, these data are obtained from clonogenic assays, and should therefore be compared to our observations using the same assay. We show IC50 values of 1.4 µM and 1.8 µM for DAOY and UW228-3 respectively in the clonogenic assay (Figure 4), two of the most resistant medulloblastoma cell lines included in the study. Patel et al, also report cell viability data from MTT assay (comparably with WST-1) for RKI-1447 in MDA-MB-231, with an IC50 value of about 10 µM (Supplementary figure S2, PMID: 22846914), a value also comparable with the more resistant medulloblastoma cell lines like DAOY.

3) Several approaches could have been considered by the authors to address this issue empirically. For example:

-Are the inhibitory effects associated with ROCK inhibitors at these concentrations specifically observed in these medulloblastoma cell lines? Or are they actually observed in any cell line? Normal fibroblast, nHDF, is mentioned in Table S1 but is not shown in Fig 2. Were ROCK inhibitors tested on nHDF cells?

-The study is solely based on the use of a pharmacological approach. Does the genetic perturbation ROCK genes (KO or overexpression) results in the same observations?

-Comparing the IC50 observed for a ROCK-WT medulloblastoma cell line in comparison to its ROCK-KO counterpart could help demonstrate the specificity of the pharmacological effect.

-How about testing the FDA-approved, selective ROCK2 inhibitor KD025?  It would be interesting to assess whether KD025 has any effect in medulloblastoma cells where ROCK2 is expressed (DAOY, UW228-3), or not (CHLA-01, MB-LU-181).

Response: We thank the Reviewer for the suggested experiments. We have added two non-tumorigenic fibroblast cell lines, nHDF (skin fibroblasts) and MRC-5 (fetal lung fibroblasts) to the cell viability evaluation of RKI-1447, AT13148 and HA1077 in the result section (page 4, line 141-), Figure 2 and Table S3. The results showed that RKI-1447, AT13848 and HA1077 have growth inhibitory effects on these non-tumorigenic fibroblast cell lines. However, the average IC50 value for the fibroblast cell lines were considerable higher compared to the medulloblastoma cell lines, with a significant difference for RKI-1447.

We have also performed knockdown experiments to study the importance of ROCK2 activity on medulloblastoma cell growth. We inhibited ROCK2 using two different siRNA pools (three siRNA sequences/each) to minimize the risk of off-targets effects from the siRNA. siRNA-mediated down-regulation of ROCK2 expression significantly suppressed the cell viability of medulloblastoma cell line D425 and DAOY after 72 h. The downregulation of ROCK2 was confirmed on protein level. These results are now included in the results section (page 6, line 205-), Figure 3B, C and Figure S2C. We choose to target ROCK2 because of the described distribution of the enzyme in the brain and the observed increased mRNA levels in metastatic samples. These results support that at least part of the observed anti-proliferative effect is caused by RKI-1447 mediated inhibition of ROCK2. However, there is still a possibility of drug-related off-target effects, especially in the higher concentrations used which is discussed in the discussion (page 12, line 385-).

We thank the reviewer for the suggestion of using K025, we however here choose to use siRNA for ROCK2 instead since there still might be off-targets effect on ROCK1 even if it’s a 200 fold change in-between the IC50s in cell-free assays for ROCK1 and ROCK2 for K025 (Boerma et al, PMID: 18832915).

Other comments:

4) In the in vivo experiment, which is limited to 10 days only, the authors observed a tumor growth suppressive effect but, in addition, they also observed a mild transient weight loss, suggesting a potential toxic effect.

Is the 10 day-limit of the experiment associated with avoiding a potentially more severe effect upon prolonged treatment?

Could this effect be attributed to the specific inhibition of ROCKs? Or could it be a non-specific toxic effect?

On the same note, the authors should consider discussing the genetic studies of ROCKs in mouse and cell and the phenotypes associated with the ROCK1/2 genetic KOs (including double KO), e.g., PMID:26765561 or PMID:30848941. In light of these observations, potential implications (side-effects?) for using ROCK inhibition as a therapeutic approach could be discussed.

Response: The 10 day-limit of the animal experiment was used in order to not exceed the maximum tumor size according to our ethical permit. This is specified in the material and method section in In vivo xenograft studies: “Duration of treatment was restricted by tumor size in the control group” (page 16, line 603-).

We cannot draw any conclusions of the mechanism behind the minor transient weight loss, it may be attributed to ROCK inhibition and vasodilation, but can also be a non-specific effect. Interfering with basic cellular functions such as cell adhesion may result in undesirable side effects. The ROCK inhibitor HA1077 (Fasudil) has been used in the clinic in Japan for a decade in treatment of vasospasm following subarachnoid haemorrhage and in clinical trials for other diagnoses in the role as a vasodilator. Fasudil is described to be safe and generally well-tolerated. However, there are reports about mild and rare side effect such as systemic hypotension, nausea, heart burn and transient headache (Hu et al, PMID: 16083339, Mueller et al, PMID: 15864268). Fasudil is usually prescribed acutely for a short time span (two weeks). The safety and efficacy profile in long-term use remains to be evaluated. We have added a section about these concerns in the discussion (page 13, line 447-)

We have also added a short discussion on effects and possible side effects based on ROCK-knockout studies. Studies have shown that in order to achieve effects on tumor cell proliferation it is preferable to target both ROCK1 and ROCK2 to avoid redundancy between the two isoforms. However targeting both ROCKs may also cause more side effects in for example cardiomyocytes. This redundancy effect of ROCK1 and ROCK2 is also emphasized on the fact that a double knockdown is lethal at an embryonic stage (page 13, line 449-).

5) Figure 1 and methods section 4.7 about the "Gene expression profiling": The authors should clarify, for each of the panel A and B,  the exact source of the data (Refs 20 only, or all four Refs 20-23) as well as the exact methods used for analysis (refering to all four Refs 20-23 is too vague), so that the readership has all the info to recapitulate the experiment.

Response: We apologize for the unclear information and have added more information and in section 4.7 in material and methods on the gene expression studies (page 16, line 591-). 

6) The authors state "ROCK2 expression was significantly higher in metastatic compared to non-metastatic samples, assessed with t-test."

A P = 0.005 on Fig. 1C is reported, demonstrating the statement, but a visual inspection of Fig. 1C does not show an obvious difference. The median values actually appear to be very similar. This visual impression may be due to the fact that only outliers data points are shown, not the bulk of them. Could the authors show all the data points maybe? In addition to the median, could the authors report the mean? Can the authors share the raw data in an excel file (normalized Affymetrix values for the non-metastatic tumors (n = 397) and metastatic tumors (n = 176)) as part of the review process. They may actually consider adding this excel file as Sup File in the manuscript?

Response: We thank the Reviewer for this suggestion and have accordingly changed Figure 1C so all data points are shown, and report both means and medians in the figure and legend (page 4, line 115-). Additionally, we have attached the data (retrieved from R2 data base) as Table S1 in an excel file.

7) Fig 4E: At the beginning of the experiment (0 h), the wound appears to be significantly bigger in the RKI condition. At 18 hours, the size of the wound appears to have been reduced essentially by ~the same length in both conditions. If the size of the wound at 0h is bigger in the RKI condition than in the control condition, then the % of area covered after 18 hours would be lower in the control condition, even if the cells were actually migrating at the same rate. Using the % of open area as a reference point is only valid if the size of the wound is similar and reproducible in all the test and control experiments.

"Four images were acquired per treatment and the experiment was repeated three-four times." For each of the 12 (or 16?) images in each of the conditions tested, what were the estimates for the surface areas (not the %) of the wound measured at 0h?

Response: We thank the Reviewer for the careful judging of the results, and apologize for choosing a not representative image in regard to wound size at time 0 h. We have according to your suggestion added information on the surface area of the wounds at 0 h in text (page 7, line 242-) and graphs shown in Figure S4. There were no differences in wound size for the three treatment groups for either DAOY or UW228-3. We have also replaced the images shown in Figure 4E to a more representative image with regard to the wound size at start.

Minor comments:

8) The authors should comment on the fact that these established cells lines are not ideal representatives of primary medulloblastoma tumors, as illustrated by clustering analysis of their genome/expression profiles in modern next-gene sequencing studies (they do not cluster with their respective medulloblastoma subgroup primary tumors). Some in the field would even argue that studies in these cells lines are not "translatable" to primary medulloblastoma biology. Hence, as a complement to their final remark ("More studies are needed to validate these results."), the authors could comment on the opportunity to test ROCK inhibitors on low-passage PDX lines.

Response: As the Reviewer correctly underline cell lines are not ideal models of primary medulloblastoma tumors, we have used one low-passage patient derived cell line MB-LU-181 (grown as neurospheres) in the manuscript, but more studies would certainly strengthen the importance of ROCK in medulloblastoma. We have added further comments on this issue in the final remarks in the manuscript (page 14, line 471-).

9) Fig 3a: total MLC at 48 hours could have been shown as well.

Response: We agree with the Reviewer and have included total MLC2 at 48 hours to Figure 3A and Figures S2B (densitometric analysis) and S8 (full blots).

Round 2

Reviewer 1 Report

The authors addressed all of the concerns. It would be interesting to see in Fig3C, how the siROCK1 and SiROCK2 affect the p-MLC-2.

Reviewer 2 Report

My comments and concerns have been addressed.